**Subject Category:**
Biology (whole organism)

behaviour

cultural evolution, frequency-based bias, music sampling, generative inference, machine learning

**Author for correspondence:**
Mason Youngblood
e-mail: myoungblood@gradcenter.cuny.edu

# Conformity bias in the cultural transmission of music sampling traditions

## Mason Youngblood[1,2]

[1]Department of Psychology, The Graduate Center, City University of New York, New York, NY, USA
[2]Department of Biology, Queens College, City University of New York, Flushing, NY, USA

MY, 0000-0003-2123-1716

One of the fundamental questions of cultural evolutionary research is how individual-level processes scale up to generate population-level patterns. Previous studies in music have revealed that frequency-based bias (e.g. conformity and novelty) drives large-scale cultural diversity in different ways across domains and levels of analysis. Music sampling is an ideal research model for this process because samples are known to be culturally transmitted between collaborating artists, and sampling events are reliably documented in online databases. The aim of the current study was to determine whether frequency-based bias has played a role in the cultural transmission of music sampling traditions, using a longitudinal dataset of sampling events across three decades. Firstly, we assessed whether turn-over rates of popular samples differ from those expected under neutral evolution. Next, we used agent-based simulations in an approximate Bayesian computation framework to infer what level of frequency-based bias likely generated the observed data. Despite anecdotal evidence of novelty bias, we found that sampling patterns at the population-level are most consistent with conformity bias. We conclude with a discussion of how counter-dominance signalling may reconcile individual cases of novelty bias with population-level conformity.

## 1. Introduction

As Darwinian approaches are increasingly incorporated into modern musicology [1], researchers have begun to investigate how transmission biases shape the cultural evolution of music [2–6]. Transmission biases, or biases in social learning that predispose individuals to favour particular cultural variants, are important selective forces [7] that can result in significant changes at the population-level [8–11]. For example, a recent study found evidence that the presence of positive and negative lyrics in popular music has been driven by prestige, success and content biases [6].

In the last several decades, researchers have begun to explore how these kinds of transmission processes can be inferred from large-scale cultural datasets. This 'meme's eye view' approach [12], originally pioneered by archaeologists studying ceramics [13,14], has since been applied to dog breeds [15], cooking ingredients [16], and baby names [17]. In music, this approach has revealed that frequency-based biases like conformity and novelty, in which the probability of adopting a variant disproportionately depends on its commonness or rarity [18], vary across domains and levels of analysis. For example, there is some evidence that dissonant intervals in Western classical music are subject to novelty bias [19], rhythms in Japanese enka music are subject to conformity bias [19], and popular music at the level of albums [15] and artists [20] is subject to random copying.[1]

Music sampling, or the use of previously recorded material in a new composition, is an ideal model for investigating frequency-based bias in the cultural evolution of music because (1) samples are known to be culturally transmitted between collaborating artists, and (2) sampling events are reliably documented in online databases [21]. For researchers, music sampling is a rare case where process is understood and pattern is accessible. In the current study, we aim to use longitudinal sampling data to determine whether frequency-based bias has played a role in the cultural transmission of music sampling traditions. Earlier manifestations of the 'meme's eye view' approach, based on diversity and progeny distributions, are time-averaged and more susceptible to type I and II error, respectively [17,22,23]. In the current study, we use two more recent methods, turn-over rates and generative inference, that better capture the temporal dynamics that result from transmission processes [24].

The turn-over rate of a top list of cultural variants, ranked by descending frequency, is simply the number of new variants that appear at each timepoint [15]. Examples of top lists in popular culture include the Billboard Hot 100 music chart and the IMDb Top 250 movies chart. By comparing the turn-over rates ($z$) of top lists of different lengths ($y$), we can gain insight into whether or not the data are consistent with neutral evolution (i.e. random copying). The turn-over profile for a particular cultural system can be described with the following function:

$$z_y = A \cdot y^x,$$ (1.1)

where $A$ is a coefficient depending on population size and $x$ indicates the level of frequency-based bias [20,25,26]. Simulation studies indicate that at neutrality $x \approx 0.86$ [20,25]. Under conformity bias turn-over rates are relatively slower for shorter top-lists, leading to a convex turn-over profile ($x > 0.86$). Likewise, under novelty bias turn-over rates are relatively faster for shorter top-lists, leading to a concave turn-over profile ($x < 0.86$) [20].

Generative inference is a powerful simulation-based method that uses agent-based modelling and approximate Bayesian computation (ABC) to infer underlying processes from observed data [27]. Agent-based modelling allows researchers to simulate a population of interacting 'agents' that culturally transmit information under certain parameters. With a single cultural transmission model, this method can be used to infer the parameter values that likely generated the observed data [23,26,28,29]. With competing models assuming different forms of bias, this method can be used to choose the model that is most consistent with the observed data [23,28,30,31]. In the current study, we use the basic rejection form of ABC for parameter inference and a random forest machine learning form of ABC for model choice.

# 2. Methods

## 2.1. Data collection

Sampling data were collected from WhoSampled (https://www.whosampled.com/) on 18 February 2019. The analysis was restricted to drum breaks because artists typically only use one drum break per composition, whereas vocal and instrumental samples are combined more flexibly. For each sample source tagged as a 'drum break', we compiled the release years and artist names for every sampling event that occurred between 1987 and 2018. Previous years had fewer than 82 cultural variants and were excluded from the analysis. Collectively, this yielded 1463 sample sources used 38 500 times by 14 387 unique artists. The release years were used to construct a frequency table in which each row is a year, each column is a sample, and each cell contains the number of times that

---

[1]Under certain conditions. The transmission of popular artists on Last.fm is consistent with random copying in generalist groups of users and conformity in more niche groups of users [20].

**Table 1.** Notable sampling events for the five most sampled drum breaks used in the current study. The number of times each drum break has been sampled was collected from WhoSampled on 27 June 2019.

| original sample | times sampled | notable sampling events |
|---|---|---|
| 'Amen, Brother' by The Winstons (1969) | 3225 | 'Straight Outta Compton' by N.W.A (1988) |
| | | 'King of the Beats' by Mantronix (1988) |
| | | 'I Want You (Forever)' by Carl Cox (1991) |
| 'Think (About It)' by Lyn Collins (1972) | 2251 | 'It Takes Two' by Rob Base & DJ E-Z Rock (1988) |
| | | 'Alright' by Janet Jackson (1989) |
| | | 'Come on My Selector' by Squarepusher (1997) |
| 'Funky Drummer' by James Brown (1970) | 1517 | 'Fight the Power' by Public Enemy (1989) |
| | | 'I Am Stretched on Your Grave' by Sinéad O'Connor (1990) |
| | | 'Pop Corn' by Caustic Window (1992) |
| 'Funky President (People It's Bad)' by James Brown (1974) | 865 | 'Eric B. Is President' by Eric B. & Rakim (1986) |
| | | 'Hip Hop Hooray' by Naughty by Nature (1993) |
| | | 'Wontime' by Smif-N-Wessun (1995) |
| 'Impeach the President' by The Honey Drippers (1973) | 785 | 'The Bridge' by MC Shan (1986) |
| | | 'Mr. Loverman' by Shabba Ranks (1992) |
| | | 'The Flute Tune' by Hidden Agenda (1995) |

particular sample was used in that year. Notable sampling events for the five most sampled drum breaks are shown in table 1, and the frequencies of 10 common and 10 rare samples through time are shown in figure 1.

## 2.2. Turn-over rates

Turn-over rates were calculated using the HERAChp.KandlerCrema package in R [26]. $x$ was calculated from top-lists up to size 142 (the minimum number of cultural variants present in a given year) across all years. The observed distribution of turn-over rates was compared to those expected under neutral conditions according to Bentley [15] and Evans & Giometto [25].

## 2.3. Agent-based modelling

Simulations were conducted using the agent-based model of cultural transmission available in the HERAChp.KandlerCrema package in R [26]. This transmission model generates a population of $N$ individuals with different cultural variants, and simulates the transmission of those variants between timepoints given a particular innovation rate ($\mu$) and level of frequency-based bias ($b$). As departures from neutrality can only be reliably detected after equilibrium has been reached, this model incorporates a warm-up period that is excluded from the rest of the analysis. Negative values of $b$ correspond to conformity bias, while positive values correspond to novelty bias. The output of this model includes turn-over rates and the Simpson's diversity index at each timepoint. Simpson's diversity index ($D$) is the probability that any two randomly selected cultural variants are of the same type, where values closer to 0 indicate high diversity and values closer to 1 indicate low diversity [32].

## 2.4. Parameter inference

Parameter inference was conducted with the rejection algorithm of ABC, using the EasyABC [33] and abc [34] packages in R, in three basic steps:

(1)  100 000 iterations of the model were run to generate simulated summary statistics for different values of $b$ within the prior distribution.

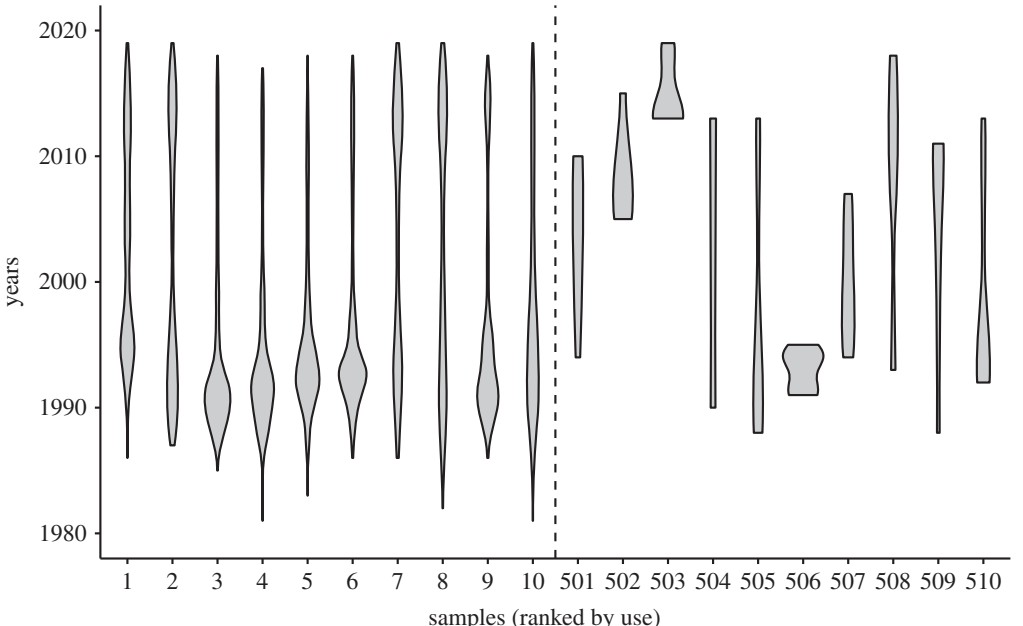

**Figure 1.** Violin plots showing the frequencies of samples, ranked by overall use, from 1980 to 2019. The x-axis is the rank of each sample, and the y-axis is the year. To the left of the dotted line are samples 1–10, while to the right are samples 501–510. More common samples (on the left) appear to be much more stable over time than rarer ones. The high popularity of the more common samples in the late 80s and early 90s is likely due to the rapid expansion of sample-based hip-hop and dance music triggered by increased access to digital samplers and more relaxed copyright enforcement during that period.

(2) The Euclidean distance between the simulated and observed summary statistics was calculated for each iteration.

(3) The 1000 iterations with the smallest distances from the observed data, determined by the tolerance level ($\varepsilon = 0.01$), were used to construct the posterior distribution of $b$.

The exponent of the turn-over function ($x$) and the mean Simpson's diversity index ($\bar{D}$) were used as summary statistics for parameter inference. Population size ($N = 729$), innovation rate ($\mu = 0.037$), and warm-up time ($t = 200$) were kept constant for all models, and a uniform prior distribution was used for $b$ ($-0.2$–$0.2$). Population size was calculated from the mean number of unique artists involved in a sampling event at each timepoint in the observed dataset. Innovation rate was calculated from the mean number of new sample types per total number of samples at each timepoint in the observed dataset, according to Shennan & Wilkinson [35]. The warm-up time was determined by running 1000 iterations of a neutral model with the observed innovation rate over 500 timepoints [23] and estimating when observed diversity reaches equilibrium (see electronic supplementary material, figure S1). The bounds of the uniform prior distribution for $b$, adapted from Crema *et al.* [23], were reduced based on observed levels of frequency-based bias in other cultural systems [26,27,29]. Each model was run for 32 timepoints, which corresponds to the number of years in the observed dataset.

## 2.5. Model choice

Model choice was conducted with the random forest algorithm of ABC, using the abcrf [36] package in R. Random forest is a form of machine learning in which a set of decision trees are trained on bootstrap samples of variables, and used to predict an outcome given certain predictors [37]. Traditional ABC methods function optimally with fewer summary statistics [38], requiring researchers to reduce the dimensionality of their data. We chose to use random forest for model choice because it appears to be robust to the number of summary statistics [36], and does not require the exclusion of potentially informative variables. The random forest algorithm of ABC was conducted with the following steps:

(1) 50 000 iterations of each model (conformity, novelty, and neutrality) were run to generate simulated summary statistics for different values of $b$ within the prior distributions.

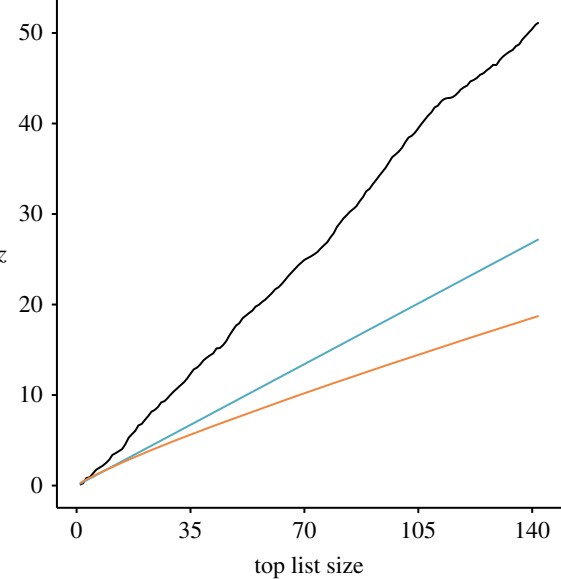

**Figure 2.** The observed turn-over rates ($z$) for top-lists up to size 142, compared to those expected under neutral conditions according to Bentley [15] (in blue) and Evans & Giometto [25] (in orange). The x-axis is the size of the top lists for which $z$, on the y-axis, was calculated.

(2) The results of these three models were combined into a reference table with the simulated summary statistics (and calculated LDA[2] axes) as predictor variables, and the model index as the outcome variable.

(3) A random forest of 1000 decision trees was trained with bootstrap samples from the reference table (150 000 rows each).

(4) The trained forest was provided with the observed summary statistics, and each decision tree voted for the model that the data were likely generated by.

(5) The posterior probability of the model with the majority of the votes was calculated using the out-of-bag data that did not make it into the bootstrap training samples.

The details of this process are outlined by Pudlo *et al.* [36]. The following 178 summary statistics were used for model choice: the exponent of the turn-over function ($x$), the mean turn-over rate ($\bar{z}_y$) for each list size (up to 142), the Simpson's diversity index for each timepoint ($D$) (up to 32), the mean Simpson's diversity index ($\bar{D}$), and the two LDA axes. Population size ($N = 729$), innovation rate ($\mu = 0.037$) and warm-up time ($t = 200$) were kept constant for all models. Uniform prior distributions were used for $b$ in both the conformity ($-0.2$–0) and novelty (0–0.2) models, whereas $b$ was kept constant at 0 for neutrality.

## 3. Results

The observed turn-over rates, as well as those expected under neutral conditions, can be seen in figure 2. Kolmogorov–Smirnov tests found that the observed distribution of turn-over rates is significantly different from the neutral expectations of both Bentley [15] ($p < 0.001$) and Evans & Giometto [25] ($p < 0.001$). The value of the exponent $x$ (see equation (1.1)) for the observed data is 1.13, which is indicative of conformity bias.

The posterior probability distribution of the level of frequency-based bias ($b$), constructed with the basic rejection algorithm of ABC, is shown in figure 3. Based on the parameter estimation of $b$, the observed data are most consistent with weak but significant conformity bias (median = −0.012; 95% HDPI: [−0.019, −0.0020]). A goodness-of-fit test ($n = 1000$; $\varepsilon = 0.01$) indicates that the model is a good fit for the data ($p = 0.47$) (see electronic supplementary material, figure S2) [39], and leave-one-out

[2]Linear discriminant analysis (LDA) is a method of dimensionality reduction, similar to PCA, that compresses multiple variables onto two axes while maximizing the separation between classes.

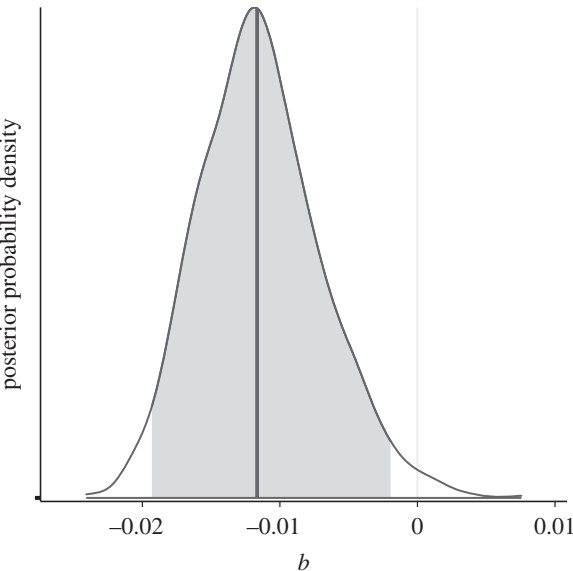

**Figure 3.** The posterior probability distribution of the level of frequency-based bias ($b$), with the median shaded in dark grey and the 95% HDPI shaded in light grey.

**Table 2.** The number of votes cast by the trained random forest for each model after being provided with the observed summary statistics, as well as the posterior probability of the selected model (conformity).

| conformity | novelty | neutrality | post. prob. |
|---|---|---|---|
| 436 | 174 | 390 | 0.89 |

cross validation indicates that the results are robust across tolerance levels ($n = 10$; $\varepsilon$: 0.005, 0.01, 0.05) (see electronic supplementary material, figure S3) [34].

The results of the model choice using the random forest algorithm of ABC can be seen in table 2. The conformity model has the strongest support (436 votes) with a posterior probability of 0.89. The out-of-bag error, calculated by running the out-of-bag data through the random forest, was 0.046 (see electronic supplementary material, figure S4), indicating that the forest is a good classifier for the data. The most important variable for the classification ability of the random forest, identified using the Gini impurity method, was mean diversity ($\bar{D}$), followed by the first LDA axis (LD1), the exponent of the turn-over function ($x$), and the second LDA axis (LD2). The importance of the top ten variables, as well as the results of the LDA, can be seen in electronic supplementary material, figures S5 and S6.

## 4. Discussion

By applying simulation-based methods to three decades of sampling events, we have provided evidence that conformity bias plays an important role in the cultural transmission of music sampling traditions. Firstly, turn-over rates for longer list sizes are higher than expected under neutral evolution, indicating that artists may be selectively using more popular samples. In addition, the rejection algorithm of ABC found that transmission models assuming low but significant levels of conformity bias best match the observed data. Lastly, a random forest trained on simulated data from three transmission models classified the observed data as coming from the conformity model. Taken together, these results indicate that music producers tend to conform to the sampling patterns of others, which is consistent with reports of artists using particular samples as signals of community membership (e.g. the Amen break) [40].

Although our results are concordant with evidence of conformity bias in Japanese enka music [19], they conflict with evidence of novelty bias in Western classical music [19] and neutral evolution in popular music [15,20]. In the study of Western classical music, frequency-based bias was identified by

looking at changes in the means and standard deviations of the frequencies of particular cultural variants [19]. Despite the fact that these measures appear to be intuitive indicators of frequency-based bias, they do not account for competition between cultural variants as frequencies change. For example, novelty bias would be expected to favour rare variants only until they become relatively common and are supplanted by rarer variants. These kinds of dynamic processes are better captured by turn-over rates and simulation-based methods. In the two studies of popular music, researchers looked at the transmission of albums and artists between listeners using data from the Billboard charts [15] and Last.fm [20]. It is possible that the low cost of listening relative to producing allows individuals to explore new music at random rather than relying on frequency-based bias. That being said, we suspect that turn-over rates on the Billboard charts [15] may not accurately reflect the behaviour of listeners, given that the charts have historically been manipulated by record labels and distributors [41]. Last.fm, on the other hand, is a more reliable source of transmission data as users can directly share music with one another in groups [20]. Interestingly, while turn-over rates of top artists in generalist groups of users were consistent with neutral evolution, rates in more niche groups (e.g. female-fronted metal) were consistent with conformity bias [20]. It is possible that individuals in more niche groups feel a greater sense of community and are influenced more by other listeners, although this idea has yet to be tested. Overall, the discrepancies between the current and previous studies indicate that frequency-based bias in music may vary depending on the level of analysis (e.g. samples between artists versus artists between listeners) and cost of adoption (e.g. work-intensive production process versus clicking a streaming link), so the results of the current study may not be generalizable across all musical domains.

A recent study on music sampling found that less popular artists culturally transmit samples with one another at higher rates [21]. In combination with anecdotes of artists selectively avoiding popular samples (e.g. De La Soul refusing to sample mainstream artists) [42], this suggests that novelty bias may be present. Counter-dominance signalling is a recently developed hypothesis [43] that may reconcile the strong conformist signal in our data with the indications of novelty bias in the literature [21,42]. This hypothesis posits that low popularity 'outsiders' develop new styles in opposition to those expressed by high popularity 'elites'. Over time, these new styles become widespread enough to be adopted by elites, allowing space for new counter-elite styles to emerge in response [43]. In other words, novelty bias may cause new styles to be adopted by outsiders, and conformity bias may allow those new styles to spread and eventually be expressed by elites. If less popular artists are much more common and tend to favour samples used within their community over those used by more popular artists, then population-level sample frequencies are likely to reflect conformity bias over novelty bias. This hypothesis is consistent with the emphasis that many artist communities place on collective cultural production in opposition to the 'mainstream' [44].

There are several limitations to this study that need to be highlighted. Firstly, the turn-over rate results should be interpreted with caution, as recent work indicates that the exponent of the turn-over function ($x$) may be overestimated when fewer than 40 timepoints are analysed [26]. Additionally, traditional ABC requires researchers to choose a subset of summary statistics, which can have a significant effect on parameter estimation. Luckily, the two statistics we used for parameter estimation ended up being the most important variables for classification by the random forest (excluding the LDA axes). Lastly, recent work indicates that the inclusion of rare variants is important for inferring underlying cultural transmission biases from population-level data [17]. As WhoSampled is a crowd-sourced database, its coverage of popular variants is presumably much more complete than its coverage of rare variants. Algorithms for sample-detection may allow researchers to reconstruct full transmission records in the future, but these approaches are not yet publicly available [45,46].

The results of the current study add to an expanding body of literature addressing how frequency-based bias influences cultural diversity at the population-level. In addition, we have provided further validation of generative inference methods that allow researchers to bridge pattern and process in cultural evolution. Future studies should employ more complex agent-based models that incorporate social status to determine whether counter-dominance signalling influences cultural transmission within music production communities, as well as other forms of transmission bias (e.g. content and prestige) to control for equifinality.

Data accessibility. All R scripts and data used in the study are available in the Harvard Dataverse repository at: https://doi.org/10.7910/DVN/TWADX4.

Competing interests. The author has declared that no competing interests exist.

Funding. The author received no specific funding for this work.

Acknowledgements. I thank all members of the Lahti laboratory for their valuable conceptual and analytical feedback.

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
