## [Reviewer comments · Royal Society Open Science]

Review History

RSOS-191149.R0 (Original submission)

Review form: Reviewer 1

Is the manuscript scientifically sound in its present form?

Yes

Are the interpretations and conclusions justified by the results?

Yes

Is the language acceptable?

Yes

Do you have any ethical concerns with this paper?

No

Have you any concerns about statistical analyses in this paper?

No

Recommendation?

Accept with minor revision (please list in comments)

Comments to the Author(s)

By analyzing drum break samples from 31 years of popular music and comparing their distribution with simulated distributions, the author provides evidence for conformity bias (as opposed to either neutral evolution or novelty bias) in the cultural evolution of music sampling.

This study tackles an interesting question of interest to many in the cultural evolution community and seems well conducted. The data sample is large and suitable for answering the question posed. The manuscript is also well written and is admirably concise.

My only real concerns are minor ones. First, I think the text could be edited and expanded in one or two places to clarify points, especially for readers less familiar with this approach. I think this study, if published, could be of interest to a rather wide range of readers, and it would be a shame if the audience is reduced. For instance, take this from the first page: "Earlier manifestations of the "meme's eye view" approach, based on diversity and progeny distributions, are time-averaged and more susceptible to type I and II error, respectively." To many readers, this will be clear. Others, however, are likely to be lost as to what diversity and progeny distributions are and why time averaging matters. On a related note, it would be helpful to clarify on page 2 how we know that x approximates 0.86 at neutrality. Simpson's Diversity Index would also benefit from being clarified in more detail (how is it calculated?).

My second concern is that the manuscript may overplay the likely generality of the findings. There is a reasonable discussion of how these findings might fit in with potentially contradictory findings using other methods and data. However, this does not address the possibility that sampling in particular may be subject to different evolutionary forces from other musical behaviors. I agree with the author that music sampling is both a very convenient focus of attention and an interesting one. However, it is also a behavior that inherently and explicitly involves a nod towards another musician. Furthermore, it is constrained by copyright law. I don't think we can take for granted, therefore, that what we learn from it can straightforwardly be applied to other aspects of musical performance. It's even possible that drum breaks might behave differently from other kinds of sampling. My point here is not that this is a problem with the study - it is reasonable to constrain the sample in this way; rather, it is simply that these questions need to be addressed and incorporated into the discussion. I would also encourage the author to establish earlier in the paper that only drum breaks were analyzed. The method mentions it in passing, as if we should already have known to expect it, and then relegates the justification for doing so to a footnote.

Finally, I spotted one typo: an absent "be" from the final line before the equation on the first page.

In summary, this is a very worthwhile and interesting study presented in a generally well written manuscript, which I consider needs only rather minor revisions to turn into a great publication.

Review form: Reviewer 2**Is the manuscript scientifically sound in its present form?**

Yes

Are the interpretations and conclusions justified by the results?

Yes

Is the language acceptable?

Yes

Do you have any ethical concerns with this paper?

No

Have you any concerns about statistical analyses in this paper?

No

Recommendation?

Accept with minor revision (please list in comments)

Comments to the Author(s)

Overall, I was very impressed by this manuscript. It has an interesting question, a simple but elegant way of answering that question, a thorough review of the relevant literature and good explanation of the findings and its limitations. I really don't have many specific suggestions to offer to improve it, just a couple of general ones. I would like to point out that I am not an expert in computational modeling so although the modeling analyses seem fine to me, it might be wise to have another reviewer with more expertise here.

First, the main limitation is simply that, although interesting, the broader implications and impact of these results are not obvious. Since that is not a criterion for RSOS, that shouldn't pose a problem for publication, but it might be nice to include a little more discussion of the broader context of why this result matters.

The other point is that, although I like the simplicity and elegance of focusing just on turnover rate as the key measured quantity, I wonder how well such a simple statistic really captures important complexities of cultural evolution. In particular, can the types of dynamic relationships discussed in the interesting "counter-dominance" model really be modeled by such a simple statistic? If not, do you have suggestions for how more complex models might approach such questions in the future?

By the way, I thought the counter-dominance model interpretation was interesting enough that it deserved to have a brief mention in the abstract.

Finally, would it be possible to add confidence intervals or other error bars to Fig. 2?

In general, though, I thought it was great and look forward to seeing it published after addressing the issues mentioned above.

Decision letter (RSOS-191149.R0)

13-Aug-2019

Dear Mr Youngblood

On behalf of the Editors, I am pleased to inform you that your Manuscript RSOS-191149 entitled "Conformity bias in the cultural transmission of music sampling traditions" has been accepted for publication in Royal Society Open Science subject to minor revision in accordance with the referee suggestions. Please find the referees' comments at the end of this email.

The reviewers and handling editors have recommended publication, but also suggest some minor revisions to your manuscript. Therefore, I invite you to respond to the comments and revise your manuscript.

- Ethics statement

- Data accessibility

<http://datadryad.org/submit?journalID=RSOS&manu=RSOS-191149>

- Competing interests

- Authors' contributions

- Acknowledgements

- Funding statement

Because the schedule for publication is very tight, it is a condition of publication that you submit the revised version of your manuscript before 22-Aug-2019. Please note that the revision deadline will expire at 00.00am on this date. If you do not think you will be able to meet this date please let me know immediately.

on behalf of Dr Alecia Carter (Associate Editor) and Kevin Padian (Subject Editor)
openscience@royalsociety.org

Associate Editor Comments to Author (Dr Alecia Carter):

I have now received two reviews of your manuscript. Both reviewers found your article to be well executed, well written and interesting, and both have provided some constructive feedback that would be useful to incorporate in a revision.

Reviewer comments to Author:
Reviewer: 1

Comments to the Author(s)

By analyzing drum break samples from 31 years of popular music and comparing their distribution with simulated distributions, the author provides evidence for conformity bias (as opposed to either neutral evolution or novelty bias) in the cultural evolution of music sampling.

This study tackles an interesting question of interest to many in the cultural evolution community and seems well conducted. The data sample is large and suitable for answering the question posed. The manuscript is also well written and is admirably concise.

My only real concerns are minor ones. First, I think the text could be edited and expanded in one or two places to clarify points, especially for readers less familiar with this approach. I think this study, if published, could be of interest to a rather wide range of readers, and it would be a shame if the audience is reduced. For instance, take this from the first page: "Earlier manifestations of the "meme's eye view" approach, based on diversity and progeny distributions, are time-averaged and more susceptible to type I and II error, respectively." To many readers,

this will be clear. Others, however, are likely to be lost as to what diversity and progeny distributions are and why time averaging matters. On a related note, it would be helpful to clarify on page 2 how we know that x approximates 0.86 at neutrality. Simpson's Diversity Index would also benefit from being clarified in more detail (how is it calculated?).

My second concern is that the manuscript may overplay the likely generality of the findings. There is a reasonable discussion of how these findings might fit in with potentially contradictory findings using other methods and data. However, this does not address the possibility that sampling in particular may be subject to different evolutionary forces from other musical behaviors. I agree with the author that music sampling is both a very convenient focus of attention and an interesting one. However, it is also a behavior that inherently and explicitly involves a nod towards another musician. Furthermore, it is constrained by copyright law. I don't think we can take for granted, therefore, that what we learn from it can straightforwardly be applied to other aspects of musical performance. It's even possible that drum breaks might behave differently from other kinds of sampling. My point here is not that this is a problem with the study – it is reasonable to constrain the sample in this way; rather, it is simply that these questions need to be addressed and incorporated into the discussion. I would also encourage the author to establish earlier in the paper that only drum breaks were analyzed. The method mentions it in passing, as if we should already have known to expect it, and then relegates the justification for doing so to a footnote.

Finally, I spotted one typo: an absent "be" from the final line before the equation on the first page.

In summary, this is a very worthwhile and interesting study presented in a generally well written manuscript, which I consider needs only rather minor revisions to turn into a great publication.

Reviewer: 2

Comments to the Author(s)

Overall, I was very impressed by this manuscript. It has an interesting question, a simple but elegant way of answering that question, a thorough review of the relevant literature and good explanation of the findings and its limitations. I really don't have many specific suggestions to offer to improve it, just a couple of general ones. I would like to point out that I am not an expert in computational modeling so although the modeling analyses seem fine to me, it might be wise to have another reviewer with more expertise here.

First, the main limitation is simply that, although interesting, the broader implications and impact of these results are not obvious. Since that is not a criterion for RSOS, that shouldn't pose a problem for publication, but it might be nice to include a little more discussion of the broader context of why this result matters.

The other point is that, although I like the simplicity and elegance of focusing just on turnover rate as the key measured quantity, I wonder how well such a simple statistic really captures important complexities of cultural evolution. In particular, can the types of dynamic relationships discussed in the interesting "counter-dominance" model really be modeled by such a simple statistic? If not, do you have suggestions for how more complex models might approach such questions in the future?

By the way, I thought the counter-dominance model interpretation was interesting enough that it deserved to have a brief mention in the abstract.

Finally, would it be possible to add confidence intervals or other error bars to Fig. 2?

In general, though, I thought it was great and look forward to seeing it published after addressing the issues mentioned above.

Author's Response to Decision Letter for (RSOS-191149.R0)

See Appendix A.

Decision letter (RSOS-191149.R1)

08-Sep-2019

Dear Mr Youngblood,

I am pleased to inform you that your manuscript entitled "Conformity bias in the cultural transmission of music sampling traditions" is now accepted for publication in Royal Society Open Science.

on behalf of Dr Alecia Carter (Associate Editor) and Kevin Padian (Subject Editor)
openscience@royalsociety.org

Appendix A

Conformity bias in the cultural transmission of music sampling traditions
Mason Youngblood
RSOS-191149

Response to Referees

Firstly, I would like to thank the editors and reviewers for taking the time to provide feedback. I have incorporated several of the suggested edits in response that I think greatly improve the quality of this manuscript. Below, I respond to each point brought up by both reviewers in italicized text.

Reviewer 1

By analyzing drum break samples from 31 years of popular music and comparing their distribution with simulated distributions, the author provides evidence for conformity bias (as opposed to either neutral evolution or novelty bias) in the cultural evolution of music sampling.

This study tackles an interesting question of interest to many in the cultural evolution community and seems well conducted. The data sample is large and suitable for answering the question posed. The manuscript is also well written and is admirably concise.

My only real concerns are minor ones. First, I think the text could be edited and expanded in one or two places to clarify points, especially for readers less familiar with this approach. I think this study, if published, could be of interest to a rather wide range of readers, and it would be a shame if the audience is reduced. For instance, take this from the first page: “Earlier manifestations of the “meme’s eye view” approach, based on diversity and progeny distributions, are time-averaged and more susceptible to type I and II error, respectively.” To many readers, this will be clear. Others, however, are likely to be lost as to what diversity and progeny distributions are and why time averaging matters. On a related note, it would be helpful to clarify on page 2 how we know that x approximates 0.86 at neutrality. Simpson’s Diversity Index would also benefit from being clarified in more detail (how is it calculated?).

Thank you for these ideas, I think expanding the writing is necessary in a couple of places. The first passage mentioned here, on “diversity and progeny distributions”, is really only relevant to expert readers in computational archaeology who may question why earlier approaches were not applied in the current study. An expanded version of that passage would likely be distracting for other readers, and would not improve their interpretation of the rest of study. For now, I am going to leave that passage as-is.

I totally agree that both the value of x at neutrality and Simpson’s diversity index need to be clarified. As such, I’ve made the following modifications. The relevant passage on the x at neutrality (one sentence after the equation in the introduction) now reads: “Simulation studies indicate that at neutrality $x \sim 0.86$ (Evans & Giometto, 2011; Acerbi & Bentley, 2014).” The relevant passage on Simpson’s diversity index (in the agent-based modeling section of the methods) now reads: “Simpson’s diversity index (D) is the probability that any two randomly selected cultural variants are of the same type, where values closer to 0 indicate high diversity and values closer to 1 indicate low diversity (Simpson, 1949).”

My second concern is that the manuscript may overplay the likely generality of the findings. There is a reasonable discussion of how these findings might fit in with potentially contradictory findings using other methods and data. However, this does not address the possibility that sampling in

particular may be subject to different evolutionary forces from other musical behaviors. I agree with the author that music sampling is both a very convenient focus of attention and an interesting one. However, it is also a behavior that inherently and explicitly involves a nod towards another musician. Furthermore, it is constrained by copyright law. I don't think we can take for granted, therefore, that what we learn from it can straightforwardly be applied to other aspects of musical performance. It's even possible that drum breaks might behave differently from other kinds of sampling. My point here is not that this is a problem with the study – it is reasonable to constrain the sample in this way; rather, it is simply that these questions need to be addressed and incorporated into the discussion. I would also encourage the author to establish earlier in the paper that only drum breaks were analyzed. The method mentions it in passing, as if we should already have known to expect it, and then relegates the justification for doing so to a footnote.

These are important concerns – Although the second paragraph of the discussion highlights how the discrepancies between this and other studies may be due to the level of analysis and cost of adoption of different musical traits, it is worth explicitly stating that the findings of the current study may not be generalizable. As such, I've appended the following onto the end of the second paragraph of the discussion: "..., so the findings of the current study may not be generalizable across all musical domains."

I agree that the use of drum breaks should be better justified in the text. I have moved the relevant footnote to the second sentence of the methods so that it is more clear.

Finally, I spotted one typo: an absent "be" from the final line before the equation on the first page.

Thank you for spotting this! I've modified it accordingly.

In summary, this is a very worthwhile and interesting study presented in a generally well written manuscript, which I consider needs only rather minor revisions to turn into a great publication.

Reviewer 2

Overall, I was very impressed by this manuscript. It has an interesting question, a simple but elegant way of answering that question, a thorough review of the relevant literature and good explanation of the findings and its limitations. I really don't have many specific suggestions to offer to improve it, just a couple of general ones. I would like to point out that I am not an expert in computational modeling so although the modeling analyses seem fine to me, it might be wise to have another reviewer with more expertise here.

First, the main limitation is simply that, although interesting, the broader implications and impact of these results are not obvious. Since that is not a criterion for RSOS, that shouldn't pose a problem for publication, but it might be nice to include a little more discussion of the broader context of why this result matters.

Although these results may reflect broader tendencies towards conformity in humans, I agree with the first reviewer that, since evolutionary forces may vary across musical domains and especially cultural domains, it is difficult to generalize the findings of the current study. For now, I think it is best to keep the current statement in the final paragraph: "The results of the current study add to an expanding body of literature addressing how frequency-based bias influences cultural diversity at the population-level".

The other point is that, although I like the simplicity and elegance of focusing just on turnover rate as the key measured quantity, I wonder how well such a simple statistic really captures important complexities of cultural evolution. In particular, can the types of dynamic relationships discussed in the interesting "counter-dominance" model really be modeled by such a simple statistic? If not, do you have suggestions for how more complex models might approach such questions in the future?

In the current study, the turn-over rate is not the key measured quantity. Rather, the exponent of the turn-over profile (which captures how turn-over rates vary with increasing top list size) is used in combination with other indicators to guide an approximate Bayesian computation algorithm. That being said, I absolutely agree that the methods in the current study are not sufficient to test hypotheses related to the counter-dominance model. Ideas for more complex analyses that may be able to test counter-dominance are proposed in the final sentence of the discussion: "Future studies should employ more complex agent-based models that incorporate social status to determine whether counter-dominance signaling influences cultural transmission within music production communities, as well as other forms of transmission bias (e.g. content and prestige) to control for equifinality."

By the way, I thought the counter-dominance model interpretation was interesting enough that it deserved to have a brief mention in the abstract.

Great idea! I have added a final sentence to the abstract with counter-dominance included.

Finally, would it be possible to add confidence intervals or other error bars to Fig. 2?

Unfortunately, as these are distributions of observed and predict turn-over rates, there are no repeated observations and thus no confidence intervals or error bars.

In general, though, I thought it was great and look forward to seeing it published after addressing the issues mentioned above.